# Constructing a Knowledge Graph from Open Statistical Data: The Case of Nova Scotia Disease Datasets

Enayat Rajabi[1], Rishi Midha[1] and Jairo Francisco de Souza[2]

[1]*1250 Grand Lake Rd., Sydney, NS, Canada*
[2]*Department of Computer Science, Federal University of Juiz de Fora, Brazil*

### Abstract

The majority of available datasets in open government data are statistical. They are widely published by different governments to be used by the public and data consumers. However, most datasets in open data portals are not provided in RDF format. Moreover, the datasets are isolated from one another while conceptually connected. Through this paper, a knowledge graph is constructed for the disease-related datasets of a Canadian government data portal, Nova Scotia Open Data. We transformed all the disease-related datasets to RDF according to the Semantic Web standards and enriched them by semantic rules and an external ontology. The ontology designed to develop the graph adheres to best practices and standards, allowing for expansion, modification and flexible re-use [1]. The study also discusses the lessons learned during the cross-dimensional knowledge graph construction and integrating open statistical datasets from multiple sources.

### Keywords

Open statistical data, Nova Scotia, Knowledge graph, Disease dataset

## 1. Introduction

The open government data movement has led to open data portals that provide a single point of access for a province or country. Open government data increases government transparency accountability, contributes to economic growth and improves administrative processes [1]. This data is published hoping that different organizations' data consumers can use it in the public and private sectors. A variety of published open datasets include multi-dimensional and statistical information such as census data, demographics, public health data (e.g., number of disease cases) [2, 3]. In itself, the data can be restrictive and not powerful enough to draw meaningful inferences. The datasets act as isolated pools of information that cannot be queried or linked. These sources are scattered in the government data portals, and users can access the information through specific searches in that data portal. The lack of meaning behind the open statistical data makes it impossible to form a network and link this kind of data to infer, create

---

[1]https://zenodo.org/record/5517236#.Ye_MsfXMJb8

*Woodstock'21: Symposium on the irreproducible science, June 07–11, 2021, Woodstock, NY*

✉ enayat_rajabi@cbu.ca (E. Rajabi); cbu19ffj@cbu.ca (R. Midha); jairo.souza@ice.ufjf.br (J. F. d. Souza)
🌐 https://erajabi.github.io/ (E. Rajabi)
🆔 0000-0002-9557-0043 (E. Rajabi)

and query knowledge [4]. Interconnectivity between isolated datasets in open data gives a machine a lot of information to work with, thereby strengthening its ability to deduce relations and infer meaning. A knowledge graph can be constructed in this study to connect various isolated datasets in open government data, and meaningful information can be inferred and queried [5]. This study focuses on constructing a knowledge graph for Nova Scotia Open Data (NSOD) disease-related datasets, a Canadian regional Open Data portal. Overall, there are 11 provinces and territories in Canada with approximately 11,771 published datasets in different domains ranging from "Business and Economy" to "Health and Wellness" in various formats (e.g., CSV, JSON, and Excel) [5]. Most of these open datasets do not allow users to export data in RDF format and are isolated while conceptually linked. Hence, a human should manually analyze the datasets to answer questions like: "Which viral diseases had the most number of cases in a province in 2017?". This study intends to answer such questions using the Semantic Web technologies such as ontologies, RDF multi-dimensional models, deductive reasoning rules, and generate a knowledge graph with semantic relationships. We link the instances of the disease-related datasets (metadata, dimensions, measures, and attributes) semantically on a schema-level following the W3C vocabularies and enrich them with a disease ontology. After constructing the knowledge graph, a quality and refinement process is performed using a specific quality metric to measure the accuracy and precision of the created knowledge graph based on existing refinement standards [6, 7].

The structure of this paper is as follows: Section 2 explains the background and the related studies in publishing datasets, particularly in the domain of multi-dimensional data. Section 3 describes the existing NSOD dataset. Section 4 presents the designed data model, ontology, and transformation process. Transformation challenges will be presented in Section 5, followed by a conclusion.

## 2. Background

A multi-dimensional structure is defined for statistical data using dimensions and measures. The literature cites many examples of researchers and organizations implementing the RDF Data Cube vocabulary for statistical data [8], [9]. As an example, [10] describes the process of improving and enriching the quality of Barcelona's official open data platform employing multi-dimensional data, applying linked open data assessment process and using external repositories as a knowledge base. In another example, [11] described how the Czech Social Security Administration (CSSA) published their official pension statistics as linked open data (LOD). These LOD datasets were modelled using the Simple Knowledge Organization System (SKOS) vocabulary and the RDF Data Cube Vocabulary. The use of open statistical data in the medical industry, in health or medical reports has been used in the literature. An ontology in the health IT interventions domain, developed and published in the study [12], builds on existing health and medical ontologies. The study outlines an inductive-deductive approach to establish a glossary, define classes and instances, and finally publish the ontology as linked open data. The PubMed knowledge graph [13] is another study in this domain created from the PubMed library. The study outlines the extraction of over 29 million records from the library to generate a graph that links bio-entities, authors, funding, affiliations and articles. Subsequent

data validation yielded promising results, and the graph can create and transfer knowledge, profile authors and organizations and realize meaningful links between bio-entities. The study covers familiar territory in terms of knowledge graph and generation compared to the work done in this research study. The use of Linked Data standards and patterns [14, 15] and strict adherence to well-established rules and protocols of the semantic Web prescribed by W3C ensure compatibility with past works as well.

## 3. Nova Scotia Open Data

Nova Scotia's government has an abundance of resources in terms of data and information, collected and stored on the Nova Scotia Open Data (NSOD) web portal [1] in the form of datasets. The main purpose of the NSOD portal is to allow individuals, particularly Nova Scotians, to efficiently access the information, understand their government, support their businesses, gain new insights, and make discoveries. The NSOD datasets are available through Socrata API[2]. In this study, we retrieved the NSOD datasets using Socrata API using the Python[3] programming language. We wrote a command-line tool to fetch the datasets and performed an exploratory analysis to understand the data. At the time of this research, there are 669 datasets in 28 categories, of which 77.8% are archived datasets, and 22.2% are currently active. The majority of the datasets were created between April 2016 and June 2016 and gradually updated each year. The majority of collected datasets were in the English language. Around 79.7% of the datasets have Nova Scotia province defined as their region, while 20.3% datasets have missing values in region metadata. The top categories of datasets are "Environment and Energy" (58), "Health and Wellness" (52), "Population and Demographics" (48), "Business and Industry" (37) and "Education" (32). Overall, we found 21 disease-related datasets in the "Health and Wellness" category by searching the NSOD web portal. Each NSOD dataset has a metadata section and an observation section that includes the statistical observations. Figure 1 shows the structure of disease-related datasets that had the same number of attributes in both metadata and observation sections. There were 13 observations in each dataset, including statistical information about disease cases in Nova Scotia between 2005 and 2017.

## 4. Methodology

A knowledge graph construction process can be performed based on the following steps: 1) Knowledge acquisition to collect semi-structured data from an API, 2) Knowledge extraction to extract entities and their relationships, 3) Knowledge fusaion: to construct an ontology, assigning entities and relationships and interlink entities to external ontologies and datasets, and 4) Knowledge storage to create knowledge graph in a triple store. To generate a knowledge graph for the disease datasets of NSOD, we follow the W3C standards to transform the ingested datasets to RDF using a data model, a custom ontology, a set of semantic rules, and an interlinking process. The following subsections describe the steps in detail.

---

[1]https://data.novascotia.ca
[2]https://dev.socrata.com/
[3]https://www.python.org

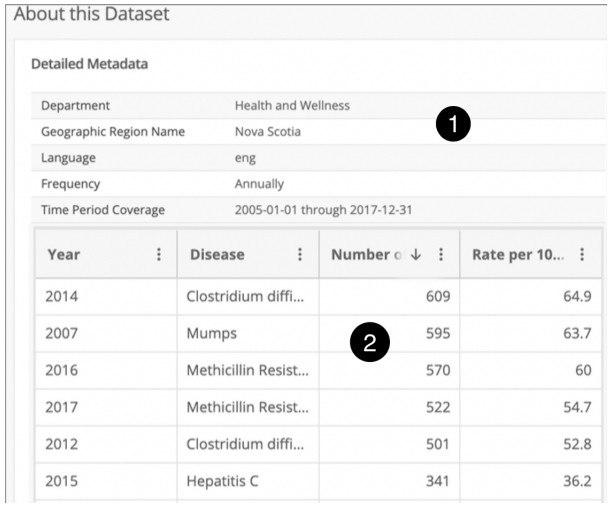

**Figure 1:** A disease dataset in the NSOD web portal (1: metadata, 2:observations)

### 4.1. Data Model

An open government dataset includes statistical information corresponding to a defined structure. The data dictionary or metadata of each NSOD dataset consists of information about that dataset, such as name, publisher, publication date, category, department, etc. which can be transformed to RDF using VoiD[16], DCMI [4], DCAT [5], and RDFS vocabularies.

The observation of an NSOD dataset includes a collection of dimensions, measures and attributes. The dimension, measures, and attributes of a dataset comprise the observation structure and are thus aptly stored in the Data Structure Definition (DSD). Figure 3 shows an example of observation in an NSOD dataset.

To model the multi-dimensional NSOD datasets, the RDF Data Cube vocabulary[6] is used based on the W3C recommendation [17]. The RDF Cube allows publishers to integrate and slice across their datasets [18] and enables the representation of the statistical data in standard RDF format and publishes the data conforming to the principles of linked data [19].

### 4.2. Ontology

To the best of our knowledge, there were no existing ontologies to be re-used based on the nature of the NSOD dataset. However, we re-used a current data model for describing multi-dimensional data (RDF Cube vocabularies), an external disease ontology (DOID), and the best practice vocabularies such as SDMX to develop a custom ontology for disease-related datasets of NSOD. The datasets were coded as entities with distinct data structure definitions, slices and observations.

All the datasets in the ontology are all instances of class *DataSet* and the nomenclature used

---

[4]https://dublincore.org
[5]https://www.w3.org/TR/vocab-dcat-2/
[6]https://www.w3.org/TR/vocab-data-cube/

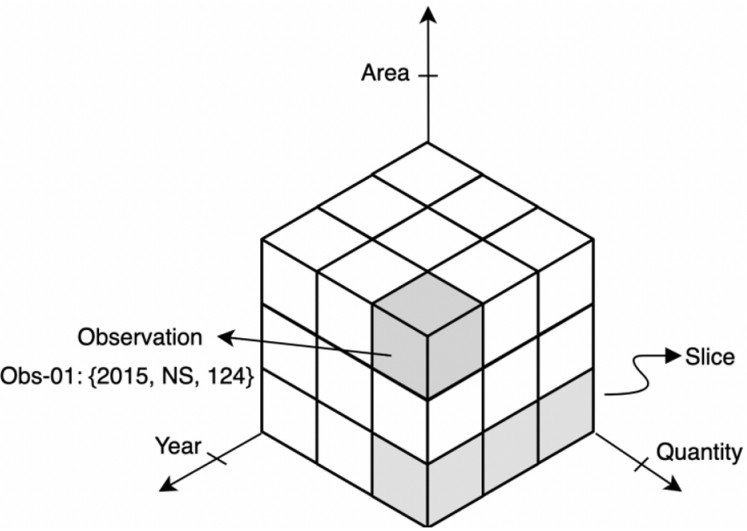

**Figure 2:** An example of observation in an open statistical dataset [5]

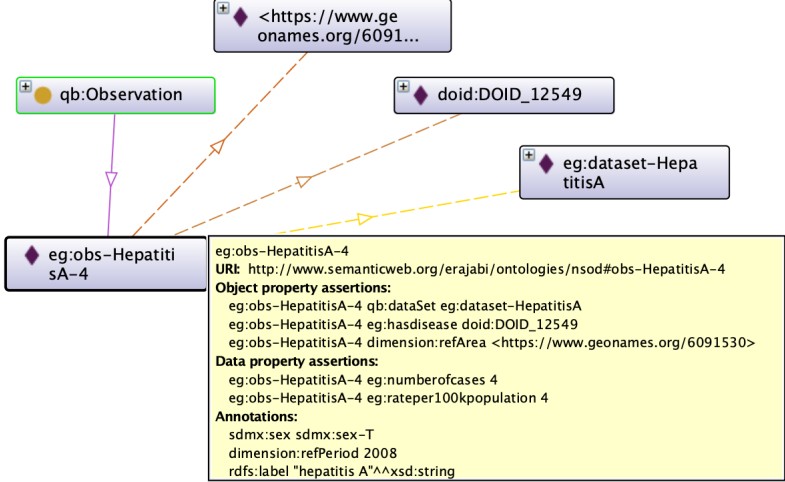

**Figure 3:** An example of observation in an open statistical dataset [5]

for datasets is *dataset-dataset_name*. Each dataset has one associated data structure definition (*DataStructureDefintion*), which defines the dataset's dimensions, measures, and attributes and is linked with *DataSet* by *structure* property. The dimensions, measures and attributes are linked with the data structure definition by properties *dimension*, *measure*, and *attribute* respectively. Also, class qb:*Slice* and *ObservationGroup* are used to group observations by one or more dimensions. Each slice is linked to the data structure definition using *sliceKey* property. The observations are attached to a dataset by the *observation* property and the respective slices by the *observationGroup* property. Figure **??** illustrates a sample observation based on the defined ontology.

**Table 1**
Re-used vocabularies

| Vocabulary | Prefix/Usage |
|---|---|
| RDF Cube | http://purl.org/linked-data/cube# |
| | Multi-dimensional, observations |
| Dublin Core | http://purl.org/dc/terms/ |
| | Metadata of datasets |
| DOID | http://purl.obolibrary.org/obo/doid# |
| | The disease ontology |
| GeoNames | http://www.geonames.org/ontology# |
| | Geographical information |
| SDMX | http://purl.org/linked-data/sdmx/2009/code# |
| | Dimensions and measures |
| SWRL | http://swrl.stanford.edu/ontologies/3.3/swrla.owl# |
| | Semantic rules |
| VoiD | http://rdfs.org/ns/void# |
| | Dataset description |

Table 1 also shows the prefixes used in the ontology.

## 4.3. Interlinking to External Ontology and Datasets

We used an external ontology (DOID[7]) to enrich the knowledge graph with domain knowledge. We imported the DOID ontology into the knowledge graph. We linked the disease name and the super-classes of each disease to the disease ontology based on the similarity of the disease names. The interlinking of datasets through their parent class is carried out, which enriches the datasets to create a sound knowledge base. We also used Geonames [8] to represent regional dimension information instead of literal adds another possibility for knowledge inference and creation. This allows the addition of semantics to statistical data in case the other datasets are joined.

## 4.4. Rules

Complex formal semantics in a knowledge graph allows a reasoner to infer the relationship between data items in different datasets [20]. This step was carried out to add more meaning to data from a dense knowledge graph and add another layer of complexity to the graph. This helps add another semantic layer and links the data together. The Semantic Web Rule Language (SWRL[9]), an example of a Rule Markup Language, was used to standardize the publishing and sharing of inference rules. As a proof of concept, we designed a SWRL rule to infer the transitive relationship of diseases in a dataset using Protege[10] rule engine. This implies that if an observation includes a disease $x$ which is a form of disease $y$ (in the disease ontology), the

---

[7]https://disease-ontology.org/

[8]https://www.geonames.org

[9]https://www.w3.org/Submission/SWRL/

[10]https://protege.stanford.edu

graph will infer that observation $x$ includes disease $y$ implicitly.

The rule states that:

$$hasdisease(?x, ?y) \quad \wedge \quad doid{:}is\_a(?y, ?z) \quad \implies \quad hasdisease(?x, ?z)$$

Another semantic rule example is related to the observations with the highest number of cases for a particular disease. Based on the current number of cases in each disease in the Nova Scotia province, we considered 1,000 disease cases per 100,000 population is high in the Nova Scotia province. Those observations are defined in the following rule:

$$Observation(?obs) \wedge numberOfCases(?obs, ?n) \wedge swrlb{:}greaterThan(?n, 1000)$$
$$\implies HighDiseaseCases(?obs)$$

The rule can be made highly specific by using constraints on threshold $N$ (number of disease cases) serving as a cut-off to classify common diseases as well as other dimensions such as region, period, gender and disease.

### 4.5. Transformation Process

A knowledge graph can be constructed in a) top-down approach where the entities are added to the knowledge-base based on a predefined ontology, or b) bottom-up approaches where knowledge instances are extracted from knowledge base systems and then, the top-level ontologies are built based on the knowledge instances to create the whole knowledge graph [21]. In this study, we followed the top-down approach to construct a disease knowledge graph from NSOD disease datasets (see Figure 4). We gathered data and transformed it into RDF triples using the designed ontology and data model described in the previous sections. The ontology was then extensively processed to enrich data through internal and external linking and dimensional and logical relations. The structural metadata about the dimensions and measures of the NSOD datasets are different in general. We developed a configuration setting to specify the dimensions and measures of each dataset, in case other datasets with various dimensions and measures are added. This allows semi-automatic updating of the graph with input data and makes the datasets semantically and dimensionally connected to the external ontologies and the Linked Open Data cloud. For example, several disease datasets had *number of cases* property that could be used as one predicate (*eg:numberOfCases*) across the knowledge graph.

In the transformation process, the Dublin Core Metadata, the most widely used metadata schema, was used to describe the metadata elements of datasets such as published date, dataset title, subject or category, source, contributor, etc. The corresponding elements of each observation were mapped to RDF triples based on the vocabularies mentioned in Table 2).

The defined rules are also translated into the constructor component to enable semantic reasoning over the knowledge graph. Finally, the datasets are added onto the graph as observations, ensuring that they conform to prescribed metadata, structure, and semantic web protocols. The graph was subjected to a quality and refinement check, and it is checked against well-received field works in terms of concept, schema, entity instances, and relations. This is followed by

**Table 2**
Mapping vocabularies

| Section | Element | Mapping voacbulary |
|---|---|---|
| Metadata | Dataset licence | dct:license |
| Metadata | Dataset language | dct:language |
| Metadata | Department | :department |
| Metadata | Dataset description | rdfs:comment |
| Metadata | Dataset keyword | dcat:keyword |
| Metadata | Dataset suject | dcat:theme |
| Observation | Year of observation | sdmx-dimension:refPeriod |
| Observation | Region of observation | sdmx-dimension:refArea |
| Observation | Number of cases for each disease | :numberOfCases |
| Observation | An observation belongs to a disease | :hasdisease |
| Observation | Case rate per 100,000 population | :rateper100kpopulation |
| Observation | Gender in observation | sdmx:sex |
| Observation | Geolocation of dataset | dct:spatial |

query retrieval to answer questions using SPARQL. The implemented Python program used for the knowledge graph construction is available at [11].

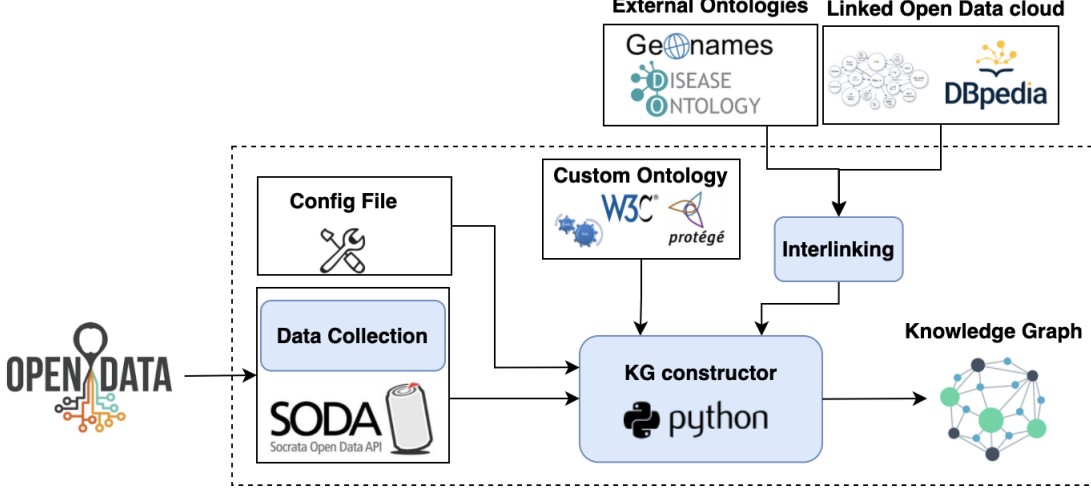

**Figure 4:** Knowledge graph construction pipeline [5]

---

[11]https://github.com/erajabi/Nova_Scotia_Open_Data

## 4.6. Queries

We used the built-in SPARQL tab in Protege to pose a set of designed queries against the knowledge base through additional semantics, which cannot be explicitly expressed through linkage. The questions were designed with the help of Nova Scotia health stakeholders. In designing the question, we considered the semantic rules developed in Section 4.4 in the knowledge graph. For example, some disease datasets were the sub-classes of the infectious disease class in the disease ontology, and we can use this property to retrieve the results. The queries are outlined below.

Figure 5 shows two questions that we defined along with the sample results. In both queries, we leveraged the rules that we defined before.

*Query 1: List of viral infectious diseases along with their number of cases in Nova Scotia in different years.*
In this query, we use *doid:is_a* relationship rule to identify all the disease classified as "viral infectious diseases".

*Query 2: List of viral infectious diseases with a high number of cases (more than 1,000 cases) in Nova Scotia in 2017.*
In this question, we use the *HighDiseaseCases* class to infer the results based upon the rule defined in Section 4.4.

**Figure 5:** Query [12]

The results of queries were cross-checked and validated for accuracy and completeness. We also performed a knowledge graph refinement process to enhance the overall quality of the knowledge graph. It includes identifying and subsequently adding the missing knowledge and correcting erroneous information. The metrics to determine the quality of a knowledge graph have been theorized based on the various refinement techniques. To determine some of these metrics, the tool OntoMetrics[13] has been utilized. The results show that the knowledge graph quality checked passed all the tests (see Table 3).

## 4.7. Knowledge Graph

The final knowledge graph included 2,883 triples with 24 classes, 23 object properties, and two data properties. All 21 disease datasets were transformed to the knowledge graph successfully with the total of 252 observation. Each observation includes Gender (*sdmx:sex*), disease information (*eg:hasdisease*), observation year (*dimension:refPeriod*), disease label (*rdfs:label*), disease rate per 100k population of disease (*eg:rateper100kpopulation*), area of observation (*dimension:refArea*) and number of disease cases (*numberofcases*) properties. The knowledge graph is publicly available at Zenodo [14] under Creative Commons Universal Public Domain Dedication (CC0

---

[12]An online SPARQL editor was used to improve the readability of the SPARQL Queries.
[13]https://ontometrics.informatik.uni-rostock.de/ontologymetrics/
[14]https://doi.org/10.5281/zenodo.5517236

**Table 3**
Quality Check Metrics With Values

| Quality Check | Description | Metric | Value |
|---|---|---|---|
| Accuracy | The correctness and validity of the information presented, verified against a legitimate source. | Spelling Error Rate | 0% |
| Domain-specificity | A horizontal or shallow ontology (high) covers more domains but not in-depth and a vertical or deep ontology (low) domain specific. | Inheritance, Richness | 77% |
| Consistency | The adherence to a structure i.e. precision. | Inconsistent,Terms Ratio | 0% |
| Informative | The information conveyed by ontology on the basis of relationships. | Relationship, Richness | 64% |

1.0)[15] license.

# 5. Conclusion and Lessons Learned

The study demonstrates the integration of disease-related datasets of an open government data portal. Due to certain limitations identified below, there is a hindrance in completing automatic constructing a knowledge graph. Although we developed a tool to retrieve open datasets from the NSOD portal, identifying the disease-related datasets was done manually, making the knowledge graph construction process semi-automatic. One of the challenges in transforming open statistical data to RDF was having different dimensions with various data types. Some of the disease-related datasets in the NSOD portal contain the same number of dimensions with the same data type, though this might not be true for all the datasets. Lack of descriptive metadata that explicitly enlist each dataset's dimensions, measures, and attributes was another significant hurdle towards achieving complete automation. Alternatively, the lack of a vocabulary that supports properties (e.g., ex:numberOfCases) that convey this information is another issue that prevents us from addressing it in a standardized manner. During the exploratory analysis of the extracted dataset, we noticed that different provincial open data portals across Canada publish datasets with the same structure and related topics. A Linked Data strategy, similar to what we described in this article, can be used to build a SPARQL endpoint (e.g., in the Canada Open Data portal [16]) to link similar open statistical datasets across a country and facilitate query answering for data consumers and the linked open data community.

---

[15]https://creativecommons.org/publicdomain/zero/1.0/
[16]https://open.canada.ca/

## 6. Acknowledgement

The work conducted in the study has been funded by the MITACS Research Training (IT21970) and NSERC (Natural Sciences and Engineering Research Council) Discovery Grant (RGPIN-2020-05869).

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
