# OpenReview forum: "Constructing a Knowledge Graph from Open Statistical Data: The Case of Nova Scotia DiseaseDatasets"
_kg-construct.github.io/KGCW/2022/Workshop — Submitted to KGCW 2022_

### Official Review · ~Manolis_Koubarakis1 · 2022-03-15
**Review for paper "Constructing a Knowledge Graph from Open Statistical Data: The Case of Nova Scotia Disease Datasets"**

**Rating:** 4
**Confidence:** 5

**Review:**

This paper presents the process followed for the construction of a knowledge graph including information about diseases using data from several disease-related datasets from the Nova Scotia Open Data portal. It is an interesting practical study but without new original
results.

Also, the paper seems not to have been proofread since there is a broken link to a figure and a missing figure.

Minor comments:
* In the introduction, start a new paragraph when you start talking about what you will do in your paper (sentence "This study focuses ...").
* Quotes come wrongly in your paper. Use `` for left quotes in LateX.
* Section 4, line 3: typo
* Last line of page 5: link missing
* page 9, Figure 5: figure missing

---

### Official Review · ~Dylan_Van_Assche1 · 2022-03-25
**A good start, but lacks a related work section**

**Rating:** 3
**Confidence:** 5

**Review:**

This paper presents the construction of a knowledge graph from Open Data of the Canadian government data portal 'Nova Scotia Open Data' around diseases. The authors tried to use as much as possible existing ontologies and practices such as Protoge which is good.
However, the paper lacks a proper related work section and does not provide any novelty to the field.

# Major comments

The authors did not provide any novelty or original results in the paper. Moreover, they seems to not fully understand the State of the Art of knowledge graph construction and did not include any related work. The authors described the knowledge graph construction process twice, once in the methodology section and once in the transformation section. Both are completely different approaches and lack some references. The authors present an approach consisting of a Python script to transform retrieved datasets into RDF which cannot be applied to other use cases in contrast to the State of the Art such as RML, xR2RML, SPARQL-Generate, etc. Several existing ontologies and a custom ontology are mentioned, but I could not understand how they are used together, an example and explanation in the paper is necessary.As already mentioned by other reviewer(s), the paper was not proofread which resulted in several typos, but also missing figures. Such figures are important since the SPARQL queries used in the paper's validation section are only visible in one of the missing figures.

In general, I miss why certain choices were made such as:
- Why a custom script for transforming these datasets into a knowledge graph?
- How the validation process was constructed and why?
- ...
Motivating why a certain choice is made is really important for others to understand why a given approach is an improvement over the State of the Art.

# Minor comments:
- Consistency: on various places, words are sometimes capitalized and sometimes not. Example: 'Semantic Web'/'semantic Web'/'semantic web'  (p3, p7)
- Examples: the readability would improve if the authors would add some examples when they say: 'we follow W3C standards'. Example: 'we follow W3C standards such as X, Y, Z' (p3).
- The last sentence of section 'Interlinking to External Ontology and Datasets' seems to have some issues and needs to be rewritten (p6).

---

### Official Review · ~Edna_Ruckhaus_Magnus2 · 2022-04-04
**Paper on a use case related to Knowledge Graoh (KG) construction. The problem is well stated and the use case is interesting but it reflects no reference to state of the art techniques on Knowledge Graph construction.**

**Rating:** 4
**Confidence:** 5

**Review:**

This paper presents a use case on Knowledge Graph construction where the source is statistical health-related open datasets for a province in Canada. It is an interesting use case and although it reuses standard vocabularies like RDF Cube for statistical data, and general and domain-specific ontologies such as the Disease ontology, it lacks reference to current well known practices on Knowledge Graph Construction. Detailed comments follow:
* The problem statement is clear.
* In Section 1, it would be clearer to extend the example on "Which viral cases...." and show which datasets need to be accessed and linked to answer this query.
* Related work makes referecnce to other use cases but techniques on KG construction are not referenced, especially the use of mapping rules is not referenced at all which would have been an alternative for this work, or if not an option, reasons should be explained.
* The details of the "KG Constructor" are not given, it is sort of a black box with no clue on the approach or techniques used.
* A good use case to look at is presented in http://vocab.ciudadesabiertas.es/def/demografia/cubo-padron-municipal/index-en.html. You can see the use of RDF Cube and the techniques that were used to generate their KG.
* Several vague and unclear sentences:
    * Section 2, "the study covers familiar territory....". What does this mean
    * Section 4.3. Why do you need to link the superclasses of each disease? What similarity measures were used?
    * Section 4.3. Whar does "in case the other datasets are joined mean?
    * Section 4.4 Why talk about a "dense knowledge graph"?
    * Section 4.4. Why do rules provide a semantic layer that "links the data together"?
    * Section 4.5 What does "the ontology was extensively processed" mean?
* Metrics using the Ontometrics tools are general ontology metrics. It is not clear how you can infer quality metrics using this tool. no details are given on the process of adding missing knowledge and correcting missing information.
* In Table 3, the relevance of the metrics Domain-specifity and Informative for this KG is not clear.
* In the ontology description some namespaces are missing. For example, ObservationGroup should be qbc:ObservationGroup.
* Should add the namespaces in Figure 3 for clarity.
* At the end of page 5, the reference to the Figure is not included (??).
* Figure 5 was not deployed.
* Zenodo link in first footnote (Page 1) is broken.
* Needs proof-reading for typos and grammar errors.

---

### Official Review · ~Mohamed_Nadjib_Mami2 · 2022-04-06
**A promising application of Semantic Technologies for real-life high-impact use case. Presentation and development is weak. Details and position to related work are missing.**

**Rating:** 4
**Confidence:** 5

**Review:**

The paper presents a study that aims at constructing a knowledge graph for Nova Scotia Open Data (NSOD) disease-related datasets, a Canadian regional Open Data portal.

Pros:
- The work is relevant and has a strong impact on its targeted domain.
- The problem is well-scoped and relevant Semantic Technologies at various levels are identified.
- A particular praise is given for clearly stating the usefulness and impact of the work.
- Source code of the implementation is made available.

Cons:
- The Background section is rather quick and shallow; describing the ways in which the presented study intersects or improves upon existing efforts is expected here. Referring to Semantic Technologies, in general, is of no use in the scope of this conference.
- The presented knowledge graph construction process is not typical, or at least partially so. Data collection is not limited to semi-structured datasets and to APIs; structured data stored as plain files or in databases is also very common. Data fusion (I guess is what you meant by the typo "fusaion") is not a typical term when referring to constructing ontologies and interlinking with existing ones. Overall, this suggests that the authors are not sufficiently versed in the State-of-the-Art of the topic.
- Describe the abbreviations and acronyms when they are first used, e.g., DOID, SDMX.
- Section 4.3 is ill-described, please proofread and clarify. Which similarity measure do you use to link the disease name to its corresponding concept in the DOID ontology? In "The interlinking of datasets through their parent class is carried out, which enriches the datasets to create a sound knowledge base.", avoid using vague terms like "carried out" and replace with the precise method used; also what does "sound knowledge base" mean? Similarly, what other possibilities are meant by "adds another possibility for knowledge inference and creation"? This continuous in the next sentence.
- Vagueness and new undefined terms like "dense" are also found in Section 4.4.
- Figure 5 is wrongly rendered (referencing a non-existing PNG file).

Minors:
- You sometimes refer to existing work [x] in the present tense (e.g., [10] describes), you other times refer to them in the past tense (e.g., [11] described).
- "The literature cites many examples of researchers and organizations...". The literature is what has been done, not citing it.
- "The use of open statistical data in ... has been used in the literature". Repetition of the verb "use".
- "To the best of our knowledge, there were no existing ontologies" -> "there are no".
- "All the datasets in the ontology are all instances" => Repeated "all".
- Overall, the present tense when describing your work (e.g., Methodology section) is preferred.
- I'd suggest consistently using camelCase format in property names, e.g., hasDisease.
- An undefined figure reference on page 5.
- "Protege" -> "Protégé".
- "knowledge-base" -> "knowledge base" (to be consistent throughout).

Despite the given comments, I would still love to see this work revised and published in the next iteration; more importantly to see Semantic Technologies driving tangible benefit in this real-life project.

---

### Decision · Program_Chairs · 2022-04-11

**Decision:**

Reject

**Comment:**

Dear authors,

Thank you for submitting your paper. Unfortunately we don’t accept your paper now in its current state. We refer to the reviews for suggestions on how you can improve your paper.

Kind regards
Organizers of the Knowledge Graph Construction workshop 2022